# Progress in the Study of Fra-2 in Respiratory Diseases

**DOI:** 10.3390/ijms25137143

**Published:** 2024-06-28

**Authors:** Shuping Zheng, Yun Liu

**Affiliations:** Department of Respiratory and Critical Care Medicine, The Second Affiliated Hospital of Xi’an Jiaotong University, Xi’an 710004, China; zsp3117331013@stu.xjtu.edu.cn

**Keywords:** Fra-2, COPD, pulmonary fibrosis, asthma

## Abstract

Fos-related antigen-2 (Fra-2) is a member of the activating protein-1 (AP-1) family of transcription factors. It is involved in controlling cell growth and differentiation by regulating the production of the extracellular matrix (ECM) and coordinating the balance of signals within and outside the cell. Fra-2 is not only closely related to bone development, metabolism, and immune system and eye development but also in the progression of respiratory conditions like lung tumors, asthma, pulmonary fibrosis, and chronic obstructive pulmonary disease (COPD). The increased expression and activation of Fra-2 in various lung diseases has been shown in several studies. However, the specific molecular mechanisms through which Fra-2 affects the development of respiratory diseases are not yet understood. The purpose of this research is to summarize and delineate advancements in the study of the involvement of transcription factor Fra-2 in disorders related to the respiratory system.

## 1. Introduction

In the late 1980s, scholars discovered the family of transcription factors known as activating protein-1 (AP-1) [1,2], and the retroviral homologues of c-Fos, which belongs to this group, were found prior to that [3]. Although the biological importance and functions of each member in the AP-1 family are still under investigation, recent advancements have shed light on how transcription factor AP-1 plays a crucial role in the regulation of cell growth, cell survival, and programmed cell death [4]. AP-1 transcription factors, which are leucine zipper proteins, include the Jun protein family (c-Jun, JunB, and JunD) [5,6], the Fos protein family (c-Fos, FosB, Fos-related antigen-1 (Fra-1, also known as Fosl-1), and Fos-related antigen-2 (Fra-2, also known as Fosl-2)) [7], the JDP protein family (JDP1 and JDP2) [8], and the ATF protein family (ATF2, LRF1/ATF3, and B-ATF) [4]. The Jun and Fos protein families constitute the majority of mammalian AP-1 transcription factors. The majority of research, both domestic and international, has concentrated on highly expressed factors such as c-Fos and c-Jun, with minimal investigation into those with lower levels of expression, such as Fra-2 and Fra-1. Research has indicated that mice lacking the *Fra-2* gene typically do not survive beyond one week after birth [9]. Consequently, studies on the role of Fra-2 in lung disease have predominantly used Fra-2 transgenic (Fra-2 Tg) mice, whereas only a few studies have used mice with tissue-specific expression of Fra-2. Nevertheless, previous studies have demonstrated that Fra-2 can impact the progression of persistent lung conditions [10] like systemic sclerosis (SSc) [11] and pulmonary fibrosis [12,13]. Research has shown that Fra-2 is crucial in controlling cell growth and differentiation, overseeing the production of the extracellular matrix (ECM), as well as maintaining the balance of signals within and outside of cells [10], but the specific molecular mechanisms of how it affects the development of respiratory diseases are still not fully understood. Recapitulating and delineating the critical function of Fra-2 in the progression of pulmonary disorders is the aim of this study.

### 1.1. Structure of AP-1/Fra-2

Basic leucine zipper (bZIP) proteins feature a crucial structural unit commonly present in transcription factors. Originally identified in research on yeast transcription factor Gen4, these proteins consist of a leucine dimerization binding domain at the C terminal and a basic DNA binding domain at the N terminal [14,15]. The binding between leucine zipper proteins is specific, and this specificity depends on the non-leucine residues of the protein [16,17]. Eukaryotic transcription factors from the AP-1 family are leucine zipper proteins that identify the TPA response element whose consensus sequence is 5′-TGAG/CTCA-3′ [1]. Additionally, they identify CRE consensus sequence 5′-TGACGTCA-3′. Jun proteins can form homodimers and heterodimers [18], whereas Fos proteins, in contrast, only create heterodimers with Jun proteins and do not form homodimers [19]. In addition to the domain that binds DNA, most AP-1 transcription factors contain a vital transactivation domain needed to start transcribing downstream target genes [19]. The arrangement of the AP-1 dimer dictates specificity for DNA binding and activation characteristics of Fra-2 [20], determining its target genes and regulatory functions.

Fra-2, although it has not been thoroughly researched yet, is a more novel member of the AP-1 family. During the 1990s, the human *Fra-2* gene was successfully cloned from a cDNA library by Matsui and colleagues [21]. Similarly, in chicken embryo fibroblasts, Nishina and colleagues [7] extracted Fra-2 and confirmed its identity as a 46-kDa protein bearing significant sequence similarity to FosB, Fra-1, and c-Fos. Subsequently, Foletta et al. [22] identified the mouse Fra-2 protein, which shows up to 87.5% homology with chicken Fra-2 and 94% homology with human Fra-2. The foundation for investigating the role of the mammalian Fra-2 protein was established. Despite having five regions similar to other Fos proteins, the Fra-2 protein distinguishes itself from c-Fos and FosB by not possessing a robust trans-activation domain (TAD) [23]. Predicting that the c-Jun/Fra-2 heterodimer may inhibit transcription, as opposed to the pairing of c-Jun and c-Fos; conversely, the Fra-2/JunD heterodimer might boost transcription when compared to the JunD homodimer [23]. These findings demonstrate the different functions of Fra-2 and indicate that AP-1 protein composition is critical for transactivation [24].

### 1.2. Expression of the Fra-2

The *Fra-2* gene can be found on chromosome 2 in humans [25]. Research has indicated that throughout human and animal growth, Fra-2 is extensively distributed in numerous tissues [26] (Figure 1). Compared to the mRNA present in other members of the Fos gene family, Fra-2 mRNA demonstrates elevated levels of expression in various adult mouse tissues, such as the heart, brain, colon, small intestine, lungs, stomach, and ovaries [22]. During the embryonic development process, Fra-2 is expressed in numerous tissues, especially during the late stages of organogenesis, as indicated by both immunohistochemistry (IHC) and in situ hybridization experiments. Fra-2 mRNA was detected in the central nervous system, differentiated epithelial cells, and developing cartilage. These findings suggest a distinct role for Fra-2 in the differentiation of cells during the growth of the fetus [26]. Previous research has indicated a possible involvement of Fra-2 in the keratinization of the skin epithelium and the formation of the endoderm in the intestine. The expression of Fra-2 in specific cell populations compared to various embryonic epithelia was demonstrated by in situ hybridization and IHC analyses of adult mouse tissues. Both granulosa cells in the ovaries and gastric pepsin-secreting principal cells exhibit expression of Fra-2 [27]. Therefore, Fra-2 might have a significant impact on cellular differentiation. Moreover, Northern blot analysis revealed that Fra-2 exhibited high levels of expression in various cancer cell lines originating from the stomach, kidneys, vulva, bladder, and breasts [21].

### 1.3. Importance of Fra-2 in Tissue Development

*Fra-2* gene-deficient mice have significant growth defects and typically do not survive beyond one week after birth [9]. The absence of Fra-2 has been shown to impact metabolism [28,29,30]; tissue development; and the differentiation of osteoblasts, chondrocytes, and osteoclasts, leading to disruptions in cartilage and bone formation [31]. Fra-2 is found in different types of epithelial cells and cartilaginous structures, playing a vital role in cartilage tissue development. This is supported by a study conducted using a Fra-2 Tg mouse model, which demonstrated an increased ability to differentiate osteoblasts and ultimately resulted in the development of osteosclerosis [32]. Research has demonstrated that Fra-2 is involved in inhibiting adiponectin production in osteoblasts, suggesting a novel mechanism by which transcription factors impact skeletal endocrine function [28]. Furthermore, it has been observed that Fra-2 is involved in regulating the growth and size of osteoclasts. Newborn mice lacking the *Fra-2* gene exhibit enlarged osteoclasts, primarily due to disrupted signaling through the leukemia inhibitory factor (LIF) pathway and its receptor [33]. Overall, Fra-2 is crucial for the regulation of chondrocytes, osteoblasts, and osteoclasts in growth and maturation processes [34].

Fra-2 is essential in influencing the body’s immune system and supporting the development of immune cells. Studies have demonstrated that the absence of Fra-2 is associated with reduced bone marrow B-cell proliferation. Further investigations involving gene analyses and chromatin immunoprecipitation (ChIP) have revealed diminished levels of Ikaros, Irf4, and Foxo1 in B cells lacking Fra-2, impacting the growth and specialization of B-cells during different stages of development [35]. A separate study demonstrated that the removal of Fra-2 led to a notable rise in peripheral and thymic iNKT cells, as well as a substantial growth in the quantity of early iNKT precursor cells. These findings indicate that Fra-2 influences the selection process of iNKT cells and is crucial for the appropriate control of iNKT cell development and functionality [36]. Ciofani et al. [37] utilized a bioinformatic approach with multiple datasets to predict the role of Fra-2 in controlling Th17 cell plasticity. Similarly, Shetty et al. [38] discovered that Th17 cell differentiation is orchestrated by a complex transcription factor network, with Fra-1 and Fra-2 involved in regulating the effector functions of human Th17 cells. Furthermore, Fra-2 is linked to the gene expression of interleukin (*IL*)-5, a crucial regulatory cytokine in eosinophilia [39]. Fra-2 is also implicated in the development of the mammalian eye. A study by McHenry et al. [40] revealed that Fra-2 Tg mice exhibited corneal abnormalities and abnormal eyelid fusion at embryonic day 15.5, causing severe disruption of eye development by 18 months of age. Researchers have also explored how Fra-2 is involved in the survival and regeneration of retinal ganglion cells after they have been removed [41]. While Fra-2 has not been implicated in light-induced photoreceptor cell death apoptosis [42], Lv et al. [42] identified its contribution to light-induced retinal injury, causing cellular demise via the PARP-1/AIF pathway.

### 1.4. Regulation of Fra-2

The interactions of AP-1 with other proteins are important in cellular biological processes [43]. AP-1 is regulated at multiple levels, the most basic of which is the distinct expression of its components. AP-1 expression is further controlled by transcription and translation, while post-translational modifications (such as phosphorylation), oncoproteins, auxiliary proteins, and other factors also impact AP-1 protein activity and stability [44]. Stimuli such as elevated cyclic adenosine monophosphate (cAMP) and Ca^2+^ levels; serum [45]; different growth factors such as transforming growth factor beta (TGF-β) [46,47]; and phorbol esters can induce transcriptional upregulation of Fra-2 (Figure 2). Research has shown that FosB and c-Fos exhibit rapid reactions to serum and maintain increased levels for a brief duration, while Fra-1 and Fra-2 display a lagged reaction but sustain heightened levels for an extended period [48]. Therefore, although multiple Fos proteins can be influenced by the same triggers, the timing of their responses varies. When induced by Ca^2+^ or cAMP, similar reactions can be observed [45]. Due to their distinct structural features and potential differential responses to external stimuli, Fra-1 and Fra-2 show greater stability than FosB and c-Fos [49]. The existence of AP-1 binding sites within the *Fra-2* promoter region implies that various AP-1 subunits engage in both autoregulatory and cross-regulatory interactions [45,50]. Specifically, the heterodimer of c-Fos/c-Jun exerts a significant stimulatory influence on the activity of the *Fra-2* promoter, leading to increased expression of Fra-2. In contrast, overexpression of Fra-2 leads to the replacement of the c-Fos/c-Jun heterodimer by the c-Jun/Fra-2 complex, resulting in reduced transcriptional activity, which indicates the presence of negative feedback regulating Fra-2 expression [23,50]. Epigenetic modifications can also impact Fra-2 expression. For example, inhibition of H3K27me3 leads to increased Fra-2 expression. Additionally, knocking down Fra-2 has been shown to completely prevent the profibrotic impacts of 3-deazaneplanocin A (DZNep). Furthermore, H3K27 histone methylation negatively regulates fibroblast activation by suppressing Fra-2 expression [51]. Like other AP-1 family members, Fra-2 undergoes transcriptional regulation and can be activated by various growth factors [47,52,53], along with inflammatory cytokines like IL-13 [54]. This activation enhances Fra-2 binding to DNA [7]. In laboratory settings, mitogen-activated protein (MAP) kinase and cAMP-dependent protein kinase A (PKA) can both phosphorylate Fra-2. However, it is only MAP kinase that induces a level and pattern of phosphorylation similar to what is observed in living organisms. MAP kinase phosphorylates Fra-1 and Fra-2, increasing their DNA-binding activity [55,56]. Conversely, the inhibition of MAP kinase ERK prevents nuclear translocation and reduces subsequent DNA-binding activity of Fra-2 [47]. Furthermore, phosphorylation contributes to the control of AP-1 protein stability and serves as a signal for degradation through the ubiquitin–proteasome pathway [57,58]. c-Jun N-terminal kinase (JNK) phosphorylates c-Jun at Ser73, protecting it from degradation and prolonging its half-life. Conversely, c-Jun was directed towards the ubiquitination pathway by phosphorylation of other N-terminal serine residues in [57]. Additionally, mitogen-activated protein kinase kinase kinase 1 (MEKK1) regulates Fra-2 protein stability and degradation [59]. Unlike other cellular regulators that are typically broken down by the proteasome after polyubiquitination, Fra-1 and c-Fos are able to be degraded independently of ubiquitination. Additionally, Fra-2 shares this unstable structural domain, indicating a possible similar mechanism for its degradation as well [60]. Previous research has demonstrated that some AP-1 family members exhibit either cooperative or opposing effects. This highlights the complex regulation of AP-1 across various stages of gene expression, encompassing transcription, translation, and protein degradation. The target genes directly regulated by Fra-2/AP-1 complexes are summarized in Table 1.

## 2. Role of Fra-2 in the Development of Respiratory Diseases

The essential role of regulating cellular behavior is played by the transcription factor known as the AP-1 complex. Dysregulation or abnormal expression of this complex can result in growth defects and the development of diseases. Proteins related to Fos, a component of the AP-1 complex, regulate various functions at different levels, impacting gene expression and ultimately affecting cell proliferation, differentiation, survival, and apoptosis [4,18,68,69]. Additionally, these proteins are involved in stress responses, organ development, immune responses, and cognitive functions [60]. Previous studies have documented that Fra-2-deficient pups have a high mortality rate shortly after birth [9,12]. Conversely, Fra2 Tg mice exhibit notable infiltration of inflammatory cells in the airways adjacent to the trachea, collagen deposition in organs, and vascular remodeling [12]. Moreover, Fra-2 is prominently expressed in SSc, a condition recognized for its connection to pulmonary hypertension and pulmonary vascular disease [46,47,67]. Multiple research studies have highlighted the increased expression and activation of Fra-2 in various lung disorders, as further elaborated in the subsequent sections.

### 2.1. Fra-2 and Chronic Obstructive Pulmonary Disease (COPD)

COPD causes high rates of mortality and disability and is a significant global health problem [70]. This condition, caused primarily by the inhalation of harmful particles, is characterized by inflammation and progressive airway obstruction. Smoking and exposure to pollutants are major contributors to the development of COPD [71]. Individuals with COPD often exhibit an increased presence of alveolar macrophages, which are pivotal in the disease process by releasing proinflammatory molecules like chemokines and cytokines [72]. Research indicates that immediate exposure to smoke extracts can enhance the expression of IL-8 and AP-1 family members such as Fra-2, Fra-1, and c-Jun. This leads to heightened production of chemokines and cytokines by macrophages, in addition to the activation of specific signaling pathways in COPD [73].

### 2.2. Fra-2 and Pulmonary Fibrosis

Pulmonary fibrosis can result from a range of lung injuries, such as toxic, autoimmune, drug-induced, infectious, and traumatic injuries. The body’s response to these injuries may be influenced by factors such as age, genetic predisposition, and environmental influences [74,75]. Fibrosis is characterized by abnormal repair processes and overproduction of the ECM, leading to the formation of scar tissue [76,77]. Studies have demonstrated that patients with pulmonary fibrosis have elevated levels of Fra-2 in their lung tissue, including both SSc-associated pulmonary fibrosis and idiopathic pulmonary fibrosis (IPF) [12,78]. This underscores the significance of Fra-2 as a crucial factor in promoting fibrosis. Upregulated Fra-2 expression in macrophages and mesenchymal cells in a mouse model of pulmonary fibrosis were found to be induced by bleomycin [10]. Targeted silencing of Fra-2 at the local level through a knockout approach resulted in improvements in fibrosis development [13]. Fra-2 exhibits pro-fibrotic activity in Fra-2 Tg mice, as found by Eferl et al. [12], particularly in the lungs and skin. This mirrors the fibrosis observed in individuals with SSc, since mouse lung tissue showed significant collagen deposition and fibrosis. Fra-2 Tg mice displayed a fibrotic phenotype similar to that seen in patients with IPF, marked by structures resembling honeycombs and the buildup of myofibroblasts synthesizing the ECM [10]. Moreover, Fra-2 Tg mice exhibited significant vascular remodeling in their pulmonary arteries, resembling the changes seen in SSc-associated pulmonary hypertension (SSc-PH) [79]. Histologically, these mice displayed characteristics typical of SSc-PH, including outer-membrane fibrosis, perivascular inflammatory infiltration, medial hypertrophy, and intimal thickening with concentric laminar lesions. Notably, there were no pulmonary occlusive venous lesions observed [11]. Pulmonary fibrotic alterations in Fra-2 Tg mice resulted in reduced inspiratory capacity and lung compliance [67]. Tabeling et al. [80] utilized the Fra-2 Tg mouse model to examine how pulmonary fibrosis affects vulnerability to Streptococcus pneumoniae and bacteremia. The presence of previously identified pulmonary fibrotic changes was confirmed by histological examination of the lungs of Fra-2 Tg mice. The presence of α-smooth-muscle actin^+^ (α-SMA^+^) cells in both the pulmonary vascular system (smooth muscle cells, SMCs) and lung parenchyma (myofibroblasts) of Fra-2 Tg mice indicates the potential involvement of Fra-2 in the growth of myofibroblasts and SMCs [10]. This suggests that airway SMCs from Fra-2 Tg mice exhibit proliferative behavior, even in the absence of growth factor activation [81]. Recently, a study questioned the idea that the specific upregulation of Fra-2 in SMCs and fibroblasts causes changes in pulmonary fibrosis and vascular remodeling, investigating whether is causes enlarged air sacs similar to emphysema [82]. Another study found that the inactivation of Fra-2 within alveolar type 2 (AT2) cells did not offer protection in a mouse model with bleomycin, suggesting that pulmonary fibrosis can still develop without Fra-2 expression in these cells [13]. In conclusion, the potential for pulmonary fibrosis resulting from elevated levels of Fra-2 in other cell types, such as macrophages, cannot be ruled out, despite fibrosis possibly being a response to address the emphysema-like characteristics observed in Fra-2 Tg mice [13]. 

In mice with Fra-2 Tg, another notable phenomenon that is worth mentioning is the infiltration of inflammatory cells, characterized by a significant accumulation of these cells in the perivascular and peribronchial regions prior to fibrosis onset [46,81]. This feature is particularly prominent in immune-mediated fibrosis, including nonspecific interstitial pneumonia (NSIP). Studies have shown that while specific cell groups (such as B cells and T cells) do not participate in the induction of fibrosis by Fra-2, other cell populations (such as macrophages) contribute significantly to the progression of fibrosis in mice [12,13]. Past research has demonstrated an increase in Fra-2 expression in pulmonary fibrosis, with its presence in alveolar macrophages being linked to the synthesis of collagen type VI (ColVI) and genes associated with M2 macrophage activation [12,13,67]. Subsequent research revealed that macrophages promote myofibroblast activation in vitro through a ColVI- and Fra-2-dependent pathway, regulate the fibrogenic function of M2 macrophages, and trigger spontaneous systemic fibrosis in Fra-2 Tg mice [13]. Research has demonstrated that the inactivation of Fra-2 in Fra-2-deficient mice or the use of a Fra-2/AP-1 inhibitor provides protection from bleomycin-induced lung fibrosis [13]. Importantly, this treatment did not affect macrophage recruitment or polarization. Therefore, while Fra-2 is pivotal in promoting fibrosis, it is not necessary for the polarization of M2 macrophages but significantly influences their fibrotic activity [13,67].

Fibrosis usually develops in various organs after ischemia/reperfusion (I/R) injury [83,84,85]. The significant involvement of Fra-2 in the formation of fibrosis induced by I/R injury has been demonstrated, particularly through its impact on the expression of TGF-β induced by oxygen in fibroblasts of the heart [65]. During fibrosis, TGF-β is recognized as a crucial element in driving the differentiation of fibroblasts and serves as a pivotal regulator in scar formation and fibrosis [86,87]. Stimulation of the TGF-β/Smad3 pathway induces the synthesis of ECM molecules, including collagen proteins like tissue inhibitor of metalloproteinase 1 (TIMP1) and collagen type VI alpha 1(Col6α1), which are known targets of this pathway [88]. On the other hand, Fra-2 is intricately associated with TGF-β signaling, as it can stimulate TGF-β expression [54] and serve as a new downstream effector of the TGF-β pathway [89,90]. TGF-β modulates the activity of the laminin alpha3A (*Lama3A*) promoter via Fra-2, as shown by Virolle et al. [66]. Additionally, it has been discovered that Fra-2 regulates autophagy and governs cardiac fibroblast differentiation through the ‘TGF-β/Fra-2/autophagy’ axis [91]. Collectively, these findings provide strong evidence for the involvement of Fra-2 in fibrosis development.

### 2.3. Fra-2 and Asthma

A common airway disease known as asthma affects many people worldwide, presenting symptoms that include difficulty breathing, tightness in the chest, coughing, and wheezing [92]. It is characterized by airway remodeling, hyper-responsiveness, and chronic inflammation. Asthma can be classified into different types, including allergic, non-allergic, exercise-induced, and cough-variant asthma [93]. Research on the role of AP-1 and Fra-2 in asthma has revealed elevated AP-1 DNA binding in bronchial fibroblasts of individuals with asthma when compared to fibroblasts from non-asthmatic controls [94]. Additionally, research has indicated that Fra-2 plays a significant part in the advancement and growth of both allergic and non-allergic asthma. Increased expression of Fra-2 in the airway tissue of individuals with allergic asthma was discovered by Huang et al. [95]. They developed a mouse model to study allergic asthma and demonstrated the activation of Fra-2 via the Janus kinase 3/signal transducer and activator of transcription 5 (JAK3/STAT5) pathway, contributing to its high expression. Subsequent in vitro experiments emphasized the involvement of M2 macrophages in driving airway inflammation and triggering allergic asthma via the JAK3/STAT5/Fra-2 pathway. During phenotypic characterization and gene expression profiling of wild-type (WT) and Fra-2 Tg mice, Gungl et al. [81] observed that Fra-2 is involved in the regulation of genes associated with airway remodeling, inflammation, and mucus secretion. The study also examined whether Fra-2 has a pronounced proinflammatory impact, potentially causing substantial Th2-induced inflammation in mouse lungs, even in the absence of extra allergen exposure. Overexpression of Fra-2 not only results in the remodeling of the pulmonary vasculature and lung parenchyma but also triggers alterations in airway/bronchial structure, leading to subepithelial fibrosis, increased airway smooth muscle thickness, peribronchial inflammatory infiltration, and airway hyper-responsiveness. Blocking IL-13 signaling or administering anti-inflammatory therapy with glucocorticoids can partially reverse this phenotype, indicating that morphological and functional changes are due to direct overexpression of Fra-2 and the activation of the IL-13 pathway [81]. Significantly, this particular characteristic is triggered without requiring external antigens, positioning it as one of the few models available for the study of endogenous asthma [81]. Fra-2 has several effects on mucus production, including inducing transcription factors such as Spdef and Foxa3, which promote goblet cell differentiation; initiating the Mucin 5AC gene (*muc5AC*); and increasing the expression of factors such as Clca1 that are important for the proper hydration and secretion of mucus [81]. Fra-2 has been shown to have a significant impact on the creation of airway mucin in smokers. This is often attributed to the heightened expression of the *Muc5AC* gene through JunD/Fra-2 dimers in bronchial epithelial cells as a result of exposure to smoke [96]. Furthermore, transcriptomic analysis revealed an upregulation of ECM genes such as collagen type I alpha 2 (*Col1α2*), collagen type VI alpha 5 (*Col6α5*), and fibronectin, along with ECM regulatory genes like matrix metalloproteinases12 (*MMP12*) and *TIMP1*, underscoring Fra-2’s role in controlling ECM deposition [47,81]. Gungl et al. [81] also found that Fra-2 has a direct proliferative effect on airway SMCs. It is probable that Fra-2 plays a crucial role in the combined effects of mucus generation, inflammatory responses, cellular growth, and collagen accumulation.

### 2.4. Fra-2 and Non-Small Cell Lung Cancer (NSCLC)

Lung cancer is the primary contributor to cancer deaths across the globe [97,98]. Approximately 85–90% of lung cancers fall under the category of NSCLC [99,100], a type of cancer that develops in epithelial cells and is known for its high metastatic capacity, making it a significant factor in mortality rates [101]. The key characteristics of aggressive and metastatic tumor growth include a transition from epithelial to mesothelial morphology and a loss of cellular polarity [102,103]. AP-1 controls the activation of numerous genes tied to cancer advancement, including those involved in genes associated with hypoxia, angiogenesis, cell differentiation and survival, invasion, and metastasis [104]. Domestic and foreign scholars have discovered that Fra-2 exhibits abnormal expression in various forms of human cancer, and this abnormal expression is known to be instrumental in the development and advancement of tumors [104,105,106,107]. This includes functions like cell proliferation, metastasis, invasion, migration, and adhesion. Research has demonstrated that Fra-2 is involved in promoting the growth of aggressive tumors such as tongue cancer [108], NSCLC [109], breast cancer [110], and thyroid-like cancer [111]. A report by Yin et al. [112] found that the mRNA expression of Fra-2 was elevated in 56 clinical NSCLC samples compared to adjacent tissues. Further research indicated a significant correlation between Fra-2 expression, tumor metastasis, and survival time. Fra-2 has also been identified as being overexpressed in squamous cell lung cancer [113]. Additionally, Fra-2 is notably expressed in breast tumor tissues and is involved in the regulation of breast cancer invasion and metastasis. Tumor cells overexpressing Fra-2 exhibited a substantial increase in proliferation, spread, metastasis, and motility [114]. A positive correlation between elevated levels of Fra-2 expression and increased recurrence rates was observed in 75 breast cancer samples [115]. The epithelial–mesenchymal transition (EMT) is aberrantly activated under pathological conditions such as cancer [116]. It plays a role in carcinogenesis by conferring an invasive phenotype and is closely associated with the separation of cancer cells of epithelial origin from the primary site and distant metastasis [117,118,119,120,121]. In vitro studies [112] utilizing a lung squamous and adenocarcinoma cell line demonstrated that overexpression of Fra-2 led to increased expression of SNAI2 (an EMT-related transcription factor [122]), promoting EMT, invasion, and migration. Conversely, knocking down Fra-2 suppressed the regulation of SNAI2 expression, invasion, EMT, and migration by HGF/MET. Subsequent research has demonstrated the crucial involvement of the HGF/MET-ERK1/2-SNAI2-Fra-2 pathway in the metastasis and EMT of NSCLC. This pathway operates by means of Fra-2 stimulating transcriptional activity by binding to the *SNAI2* promoter, thereby facilitating SNAI2 transcription. Furthermore, Fra-2 plays a crucial role in facilitating HGF/MET-triggered EMT, invasion, and migration. In their study, Song and colleagues [123] discovered that hsa_circ_0003998 enhances EMT in hepatocellular carcinoma (HCC) through its role as a sponge for miR-143-3p, leading to the up-regulation of Fra-2 expression. Additionally, a separate study [124] revealed that increased levels of Fra-2 impact the metastatic potential of cancer cells through the modulation of adhesion molecule expression. The pathway of TGF-β controls a number of crucial biological processes essential for cancer advancement and has a significant impact on various cellular activities, including proliferation, differentiation, blood vessel formation, immune system reaction, programmed cell death, cellular attachment, and cellular movement [125,126,127]. A newly discovered controller of TGF-β, Fra-2, is instrumental in overseeing the TGF-β pathway [109]. Fra-2 is involved in matrix remodeling and fibrosis as a unique subsequent mediator of TGF-β [47] while also serving as a transcriptional regulator that promotes TGF-β expression [65]. Fra-2 triggers the expression of lysine oxidase-like 4 (LOXL4) induced by TGF-β, which is vital for the control of ECM synthesis and remodeling [89]. Furthermore, the association between Fra-2 and TGF-β pathways during tumor development is evident, with TGF-β playing a role in inducing Fra-2 expression and Fra-2 serving as a novel mediator in the TGF-β pathway. A study in NSCLC highlighted the control of Fra-2 expression by TGF-β1 and elucidated its involvement in promoting TGF-β1-mediated migration of NSCLC cells via its association with Smad3 [109]. Subsequent research demonstrated that Fra-2 facilitates the interaction between P300 and Smad3, leading to the acetylation of Smad3 by P300. This suggests that Fra-2 actively participates in regulating TGF-β signaling. Clinical investigations have revealed a correlation between Fra-2 expression in cancerous tissues and postoperative recurrence, as well as survival rates in lung cancer patients [109]. Furthermore, TGF-β stimulation induces the upregulation of Fra-2 and Smad3 in HCC [109,114]. In summary, Fra-2 enhances the invasive capabilities of malignant tumors through the TGF-β pathway. The accumulation of tumor-associated macrophages (TAMs) in lung cancer is closely linked to a negative prognosis and overall patient survival [128]. Interactions between TAMs and tumor cells stimulate the production of cytokines and chemokines, which promote excessive cell growth and metastasis in lung cancer and related conditions [129,130]. Research has shown that the activation of Fra-2 and the repression of ARID5A through β-catenin-mediated transcription are significant factors in the transformation of macrophages from M1-like to M2-like TAMs in lung cancer [131]. Furthermore, hsa_circ_0001869 has been identified to enhance the expression of Fra-2 in NSCLC tissues by interacting with miR-638, resulting in increased proliferation, migration, and invasion in NSCLC [132]. Overall, the excessive expression of Fra-2 is closely associated with the growth, progression, and prognosis of malignant tumors. The critical involvement of Fra-2 expression in osteoblasts in controlling the bone marrow microenvironment and enhancing osteopontin (OPN) expression has been demonstrated [61]. This process promotes the development of mesenchymal stem cells (MSCs) and governs the movement of myeloid cells into the bloodstream. These effects subsequently induce a systemic inflammatory response, leading to heightened inflammation in the lungs. In chronic inflammatory conditions, the abnormal expression of Fra-2 and other AP-1 molecules can elevate the expression of inducible costimulatory molecule (ICOS) in T cells, contributing to disease progression [133]. Renoux and colleagues [134] discovered that Fra-2 Tg mice displayed an inflammatory profile across various organs, marked by immune cell infiltration. They also observed that Fra-2 led to decreased T-cell-intrinsic Treg formation correlated with an inflammatory characteristic. These findings indicate that Fra-2 hinders Treg development, leading to the promotion of autoimmunity and inflammation.

In conclusion, Fra-2 is crucial not only in the progression of respiratory conditions (Table 2) like COPD, pulmonary fibrosis, asthma, and lung tumors but also in various other diseases, including cardiovascular issues [135,136,137], inflammatory disorders, and autoimmune conditions. Additionally, Fra-2 plays a significant role in physiological processes like alveolar regeneration and epidermal differentiation. The ability of adult stem cells to acquire cellular plasticity is essential for prompt tissue regeneration following injury. Choi et al. [138] discovered that in the process of alveolar regeneration, IL-1β signaling regulates Jag1/2 expression in ciliated cells, which then inhibit Notch signaling in secretory cells. This pathway is responsible for driving reprogramming and facilitating differentiation plasticity. Subsequent research revealed that the IL-1β–Notch–Fra-2 pathway is crucial in the transformation of secretory cells into AT2 cells in the process of repairing damage.

## 3. Characterization and Limitations of the Fra-2 Tg Mouse Model

Fra-2 Tg mice display various characteristics of peripheral vascular lesions seen in human SSc [139]. Apoptosis of dermal endothelial cells (ECs) in 9-week-old Fra-2 Tg mice occurred significantly before the development of microangiopathy and skin fibrosis [139,140]. This finding implies that, like in human SSc, EC apoptosis could be the primary pathogenic event leading to microangiopathy in this particular model. Capillary density in Fra-2 Tg mice was similar to that of WT mice at 9 weeks of age but decreased significantly starting from week 12 [139]. At 9 weeks of age, Fra-2 Tg mice exhibited elevated perivascular inflammatory infiltration in the skin, resembling another characteristic of early human SSc [139]. Furthermore, starting from week 12, Fra-2 Tg mice displayed a progressive increase in skin thickness attributed to ECM accumulation [139]. However, peripheral microangiopathy (such as ulcers or tissue necrosis) was not clinically evident in Fra-2 Tg mice during the 16-week observation period. In addition to peripheral lesions, internal organs, particularly the lungs, were impacted in Fra-2 Tg mice [12]. Unlike dermal fibrosis, there was no observed increase in apoptosis in the lungs [78]. By 12 weeks, Fra-2 Tg mice exhibited pulmonary artery occlusion accompanied by perivascular inflammatory infiltrates, occurring 2–3 weeks earlier than the onset of fibrosis [78]. Subsequent stages revealed a significant rise in interstitial inflammation and fibrosis [78]. Ultimately, the lungs of Fra-2 Tg mice exhibited histological features commonly observed in human NSIP and IPF [78]. These features encompassed fibrosis, honeycombing, and dense lymphocytic infiltration. However, in another study by Maurer et al. [11], the phenotype of interstitial lung disease (ILD) in Fra-2 Tg mice was similar to that of human NSIP, whereas fibroblast foci and honeycombing associated with usual interstitial pneumonitis (UIP) were rarely seen. Inflammation is not a prominent feature of human IPF but is more pronounced in immune-mediated and idiopathic cases of NSIP. Perivascular and peribronchial inflammatory infiltrates were present in the lungs of fra-2 Tg mice; however, subsequent studies have indicated that T and B lymphocytes are not essential for the development of lung fibrosis [12]. Increased vessel wall thickness and pulmonary artery occlusion are the other most prominent features of lung pathology in Fra-2 Tg mice [11]. These features include intimal thickening with predominantly concentric laminar lesions, mesangial hypertrophy, perivascular inflammatory infiltration, epicardial fibrosis, and pulmonary fibrosis with interstitial inflammatory infiltration [11]. However, Fra-2 Tg mice do not exhibit pulmonary occlusive venopathy, unlike human SSc-PH. Additionally, these mice do not display the intricate lesions, such as plexiform and thrombotic lesions, that are typically observed in idiopathic pulmonary arterial hypertension (IPAH). Concentric and eccentric nonlaminar lesions are also rarely detected in Fra-2 Tg mice. In summary, the Fra-2 Tg mouse model recapitulates many aspects of human lung pathology, including inflammation, vascular remodeling, and fibrosis, but does not recapitulate all the abnormalities of human disease. 

Nevertheless, the Fra-2 Tg mouse model can serve as a valuable model to elucidate key pathological mechanisms of lung remodeling and to identify potential therapeutic approaches. The advantage of the Fra-2 Tg mouse model lies in its capacity to spontaneously develop phenotypes without the need for disease induction. This model effectively replicates several elements of human SSc-ILD pathology, such as inflammation, vascular remodeling, and lung fibrosis. The complexity of phenotypes and systemic implications in this mouse model closely resemble those observed in humans, thereby aiding in the validation of potential therapeutic targets. The Fra-2 Tg mouse model presents a limitation due to the widespread overexpression of Fra-2 across various cell types, leading to a complex phenotype that impacts multiple tissues. This lack of cell specificity raises uncertainty regarding whether the pathological effects of Fra-2 are directly or indirectly influenced by its presence. Moreover, the high phenotypic variability observed in Fra-2 Tg mice may be attributed to modulation of the inserted transgene by the genetic environment. Finally, because Fra-2 Tg mice do not develop autoimmunity, they can only be considered a model of the microvascular and fibrotic disease manifestations of SSc. The lack of the *Fra-2* gene in mice is lethal, and most studies to date on the role of Fra-2 in lung disease have used Fra-2 Tg mice, with only a few studies using tissue-specific expression of Fra-2. However, with the development of science and technology, the selection of tissue-specific Fra-2 knockout/overexpressing mice in future experiments may avoid some of the limitations of the Fra-2 Tg mouse model.

## 4. Summary and Outlook

During human and animal development, Fra-2 is widely expressed in numerous tissues. Fra-2 is involved in regulating cellular behavior, and dysregulation or abnormal expression can result in growth defects and disease development. Its main functions include controlling proliferation, differentiation, apoptosis, and stress response and playing crucial roles in organ development, immune response, cognitive function, respiratory diseases, cardiovascular diseases, and other processes. Despite its importance, the exact molecular mechanisms of how Fra-2 affects disease progression remain unclear. Regulation of Fra-2 expression involves transcription, translation, posttranslational modifications, growth factors, proinflammatory cytokines, and potentially epigenetic modifications. The exact mechanisms governing Fra-2 regulation in pulmonary fibrosis, tumors, and autoimmune diseases remain unclear. Additional research is necessary to elucidate the regulatory pathways of abnormal Fra-2 expression in disease progression and to enhance our understanding of how Fra-2 contributes to disease pathogenesis. Exploring the mode of action of Fra-2 and its specific molecular pathways in respiratory disease progression, as well as understanding the regulatory mechanisms controlled by Fra-2 during disease advancement, could facilitate the development of novel therapeutic approaches. Recent studies suggest that AP-1 inhibitors have shown promise in improving outcomes in various pulmonary fibrosis models, making AP-1 inhibition a novel therapeutic focus for the treatment of pulmonary fibrosis (Table 3). However, targeting Fra-2 directly may be challenging due to its multifaceted actions. It remains to be seen whether the hypothesized drugs (diethylstilbestrol and D-leucine) and compounds (N-acetylcysteine and PD98059) are effective against Fra-2 (Table 3).

## Figures and Tables

**Figure 1 ijms-25-07143-f001:**
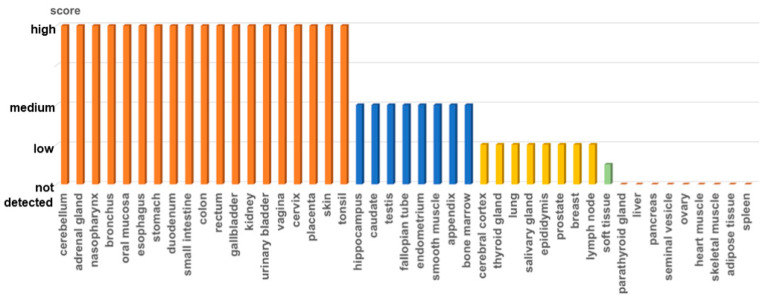
Fra-2 protein expression in the human body. Data on Fra-2 protein expression in different organs of the human body were obtained from the Human Protein Atlas database (https://www.proteinatlas.org/, accessed on 20 June 2024). This figure was plotted using Excel. Abbreviations: Fra-2, Fos-related antigen-2.

**Figure 2 ijms-25-07143-f002:**
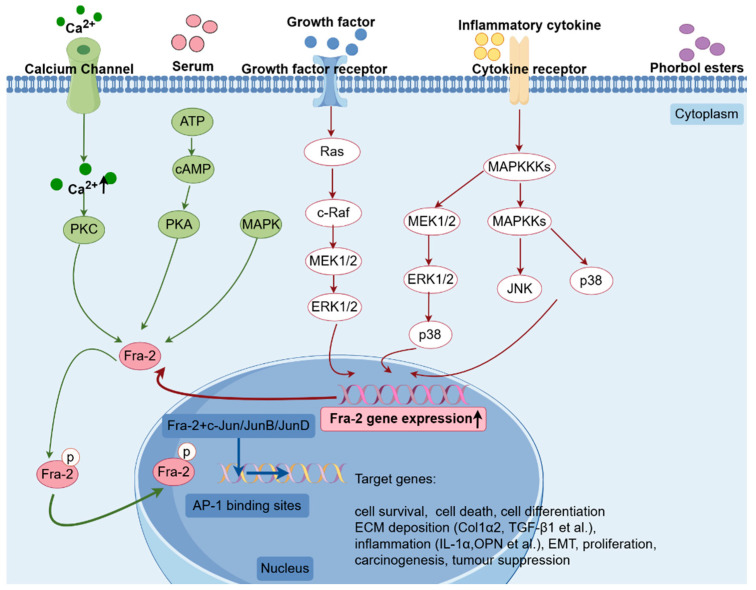
Regulation of Fra-2 and the role of downstream target genes. Fra-2 forms a dimer with c-Jun/JunB/JunD that binds to the AP-1 consensus site of target genes and induces transcription of target genes involved in a variety of processes, including ECM deposition and inflammation. Abbreviations: MEK, mitogen-activated protein kinase kinase; MAPKKK, mitogen-activated protein kinase kinase kinase; MAPKK, mitogen-activated protein kinase kinase; MAPK, mitogen-activated protein kinase; ATP, adenosine triphosphate; cAMP, cyclic adenosine monophosphate; JNK, c-Jun N-terminal kinase; PKA, protein kinase A; PKC, protein kinase C; AP-1, activating protein-1; Fra-2, Fos-related antigen-2; ECM, extracellular matrix; Col1α2, collagen type I alpha 2 chain; TGF-β1, transforming growth factor beta 1; IL-1α, interleukin 1 alpha; OPN, osteopontin; EMT, epithelial–mesenchymal transition. This image was drawn by Figdraw2.0.

**Table 1 ijms-25-07143-t001:** Known direct target genes of Fra-2.

Gene	Full Name	Details
*OPN*	Osteopontin	Luo et al. [61] validated this by ChIP and luciferase reporter assay in mouse osteoblasts.
*Fra-2*	Fos-related antigen-2	Davies et al. [62] validated these by ChIP and electrophoretic mobility shift assay (EMSA) in rat pineal body tissue.
*Rgs4*	Regulator of G protein signaling 4	
*Atf4*	Activating transcription factor 4	
*Cox6a2*	Cytochrome c oxidase subunit 6A2	
*Nr4a1*	Nuclear receptor subfamily 4 group A member 1	
*Mt1a*	Metallothionein 1 A	
*Opn1sw*	Opsin 1, short-wave sensitive	
*Dio2*	Iodothyronine deiodinase 2	
*CD24*	CD24 molecule	
*Fra-1*	Fos-related antigen-1	Adiseshaiah et al. [63] validated this by ChIP in A549 cells.
*CCND1*	Cyclin D1	Bakiri et al. [64] validated these by luciferase reporter assay and EMSA in NIH 3T3 cells.
*CCNA2*	Cyclin A2	
*TGF-β*1	Transforming growth factor beta 1	Fichtner-Feigl et al. [54] validated this by luciferase reporter assay and EMSA in THP-1 and MonoMac6 cells. Roy et al. [65] validated this by luciferase reporter assay in mouse cardiac fibroblasts.
*Lama3A*	Laminin alpha3A	Virolle et al. [66] validated this by β-galactosidase assay and EMSA in PAM212 cells.
*Col1α2*	Collagen type I alpha 2 chain	Bozec et al. [32] validated this by ChIP and luciferase reporter assay in mouse osteoblasts.
*Oc*	Osteocalcin	
*IL-1α*	Interleukin 1 alpha	Birnhuber et al. [67] validated this by EMSA in primary human parenchymal fibroblasts and mice.

**Abbreviations:** ChIP: chromatin immunoprecipitation; EMSA, electrophoretic mobility shift assay.

**Table 2 ijms-25-07143-t002:** Fra-2 expression in respiratory diseases.

Respiratory Diseases	Regulation of Fra-2	Reference
COPD	Up-regulation	Kent et al. [73]
Pulmonary fibrosis	Up-regulation	Eferl et al. [12] Ucero et al. [13] Birnhuber et al. [67]
SSc-PH	Up-regulation	Maurer et al. [11] Biasin et al. [46]
Asthma	Up-regulation	Gungl et al. [81] Huang et al. [95]
NSCLC	Up-regulation	Wang et al. [109] Yin et al. [112] Gao et al. [113] Sarode et al. [131] Xu et al. [132]

**Abbreviations:** Fra-2: Fos-related antigen-2; COPD: chronic obstructive pulmonary disease; SSc-PH: SSc-associated pulmonary hypertension; NSCLC: non-small cell lung cancer.

**Table 3 ijms-25-07143-t003:** Drugs and compounds for Fra-2/AP-1.

Inferred Drugs for *Fra-2* Gene		
Name	Status	Mechanism of Action
Diethylstilbestrol	Approved, investigational, withdrawn	Small molecule, hormone replacement agents
D-leucine	Experimental, investigational	-
**Compounds for Fra-2/AP-1**		
**Name**	**Reference**	
T-5224 (AP-1 inhibition)	Ucero et al. [13]	
N-acetylcysteine	Li et al. [141]	
PD98059	Won et al. [142]	

Data for inferred drugs for the *Fra-2* gene (diethylstilbestrol and D-leucine) and additional inferred compounds for Fra-2 (N-acetylcysteine and PD98059) were obtained from the GeneCard database (https://www.genecards.org/, accessed on 20 June 2024). **Abbreviations:** AP-1: activating protein-1; Fra-2: Fos-related antigen-2.

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
