# Peer review of "Progress in the Study of Fra-2 in Respiratory Diseases"

_ijms, 2024, doi:10.3390/ijms25137143_

Round 1

Reviewer 1 Report

Comments and Suggestions for Authors

Zheng S et al present a review on the role of Fra-2, a member of the AP-1 transcription factor family, in respiratory diseases. The objective of this review was clearly to put together most of the findings that have been published on this topic during the last years and to provide a comprehensive survey of the recent lung pathologies involving Fra2 or AP1. In its present form, the paper succeeds in bringing together most of the recent contributions without attempting to simplify and to fit a single hypothesis.

The task, however, is difficult since other AP1 members have been shown to affect many types of lung diseases, and that Fra2 is probably one lesser studied member of the AP1 family.

General comments. The authors should mention that there are important limitations in the studies aimed at defining the role of Fra2. Indeed, Fra2 gene deficiency is lethal in mice, and therefore, most of the information about Fra2 in lung diseases comes from Fra2 overexpressing mice. These mice were first described in 2008. Among mouse studies, only a few have been devoted to tissue specific expression of Fra2. Thus, a first para about the methodology used to study Fra2 would be important to emphasize the methodological limitations

Moreover, it would be important to report how we progressed in understanding the role of Fra2. For example, Fra‑2/TG mice were initially presented as an original model linking pulmonary vascular remodeling and fibrogenesis. Further characterization of these mice showed that the severe loss of capillaries in the skin was preceded by endothelial-cell (EC) apoptosis and, at 12 weeks, the development of progressive skin fibrosis. Thus, Fra-2/TG mice are now used as a model of systemic sclerosis-associated interstitial lung disease, which also should be discussed since the model does not recapitulate all the abnormalities of the human disease.

Only a few studies have been devoted to tissue specific expression of Fra2 in mice, the authors should state what we learned from these models.

In general, the authors should be more critical to their conclusions. Showing that Fra2 is overexpressed in a given condition does not mean that it plays an important role in this condition.

Also, since Fra2 is considered as an important transcription factor, providing examples of genes whose expression in regulated by Fra2 would be important to comment.

Author Response

Response to Reviewer 1 Comments

Dear reviewer,

Thank you very much for your comments and professional advice. These opinions help to improve academic rigor of our article. Based on your suggestion and request, we have made corrected modifications on the revised manuscript. Meanwhile, the manuscript had be reviewed and edited by language services of AJE. We hope that our work can be improved again. Furthermore, we would like to show the details as follows:

Reviewer 1

Comments and Suggestions for Authors:

Zheng S et al present a review on the role of Fra-2, a member of the AP-1 transcription factor family, in respiratory diseases. The objective of this review was clearly to put together most of the findings that have been published on this topic during the last years and to provide a comprehensive survey of the recent lung pathologies involving Fra2 or AP1. In its present form, the paper succeeds in bringing together most of the recent contributions without attempting to simplify and to fit a single hypothesis.

The author’s answer: Thank you very much for your suggestion. We agree with this suggestion and have carefully drawn 2 diagrams and 3 tables to categorise and simplify the content of the article for the convenience of our readers.

The task, however, is difficult since other AP1 members have been shown to affect many types of lung diseases, and that Fra2 is probably one lesser studied member of the AP1 family.

General comments. The authors should mention that there are important limitations in the studies aimed at defining the role of Fra2. Indeed, Fra2 gene deficiency is lethal in mice, and therefore, most of the information about Fra2 in lung diseases comes from Fra2 overexpressing mice. These mice were first described in 2008. Among mouse studies, only a few have been devoted to tissue specific expression of Fra2. Thus, a first para about the methodology used to study Fra2 would be important to emphasize the methodological limitations.

The author’s answer: Thank you very much for your suggestion. We agree with this suggestion and have added ‘Research has indicated that mice lacking the Fra-2 gene typically do not survive beyond one week after birth [9]. Consequently, studies on the role of Fra-2 in lung disease have predominantly used Fra-2 Tg mice, whereas only a few studies have used mice with tissue specific expression of Fra-2.’ to the ‘1. Introduction’ section of the article, which is highlighted in red in the text.

Moreover, it would be important to report how we progressed in understanding the role of Fra2. For example, Fra‑2/TG mice were initially presented as an original model linking pulmonary vascular remodeling and fibrogenesis. Further characterization of these mice showed that the severe loss of capillaries in the skin was preceded by endothelial-cell (EC) apoptosis and, at 12 weeks, the development of progressive skin fibrosis. Thus, Fra-2/TG mice are now used as a model of systemic sclerosis-associated interstitial lung disease, which also should be discussed since the model does not recapitulate all the abnormalities of the human disease.

Only a few studies have been devoted to tissue specific expression of Fra2 in mice, the authors should state what we learned from these models.

The author’s answer: Thank you very much for your suggestion. We agree with this suggestion and have added a third subsection to the paper to elaborate on the disease progression of the Fra-2 Tg mouse model, the similarities and differences with human disease abnormalities, and discuss the advantages and disadvantages of the Fra-2 Tg mouse model. Details are as follows:

  1. Characterization and limitations of the Fra-2 Tg mouse model

Fra-2 Tg mice display various characteristics of peripheral vascular lesions seen in human SSc [140]. Apoptosis of dermal endothelial cells (EC) in 9-week-old Fra-2 Tg mice occurred significantly before the development of microangiopathy and skin fibrosis [140, 141]. This finding implies that, like in human SSc, EC apoptosis could be the primary pathogenic event leading to microangiopathy in this particular model. Capillary density in Fra-2 Tg mice was similar to that of WT mice at 9 weeks of age, but decreased significantly starting from week 12 [140]. At 9 weeks of age, Fra-2 Tg mice exhibited an elevated perivascular inflammatory infiltrate in the skin, resembling another characteristic of early human SSc [140]. Furthermore, starting from week 12, Fra-2 Tg mice displayed a progressive increase in skin thickness attributed to extracellular matrix accumulation [140]. However, peripheral microangiopathy (such as ulcers or tissue necrosis) was not clinically evident in Fra-2 Tg mice during the 16-week observation period. In addition to peripheral lesions, internal organs, particularly the lungs, were impacted in Fra-2 Tg mice [12]. Unlike dermal fibrosis, there was no observed increase in apoptosis in the lungs [78]. By 12 weeks, Fra-2 Tg mice exhibited pulmonary artery occlusion accompanied by perivascular inflammatory infiltrates, occurring 2-3 weeks earlier than the onset of fibrosis [78]. Subsequent stages revealed a significant rise in interstitial inflammation and fibrosis [78]. Ultimately, the lungs of Fra-2 Tg mice exhibited histological features commonly observed in human NSIP and IPF [78]. These features encompassed fibrosis, honeycombing, and dense lymphocytic infiltration. However, in other study by Maurer et al [11], the phenotype of interstitial lung disease (ILD) in Fra-2 Tg mice was similar to that of human NSIP, whereas fibroblast foci and honeycombing associated with usual interstitial pneumonitis (UIP) were rarely seen. Inflammation is not a prominent feature of human IPF, but is more pronounced in immune-mediated and idiopathic cases of NSIP. Perivascular and peribronchial inflammatory infiltrates were present in the lungs of fra-2 Tg mice; however, subsequent studies have indicated that T and B lymphocytes are not essential for the development of lung fibrosis [12]. Increased vessel wall thickness and pulmonary artery occlusion are the other most prominent features of lung pathology in Fra-2 Tg mice [11]. These features include intimal thickening with predominantly concentric laminar lesions, mesangial hypertrophy, perivascular inflammatory infiltrate, epicardial fibrosis and pulmonary fibrosis with interstitial inflammatory infiltrate [11]. However, Fra-2 Tg mice do not exhibit pulmonary occlusive venopathy, unlike human SSc-PH. Additionally, these mice do not display the intricate lesions, such as plexiform and thrombotic lesions, that are typically observed in idiopathic pulmonary arterial hypertension (IPAH). Concentric and eccentric nonlaminar lesions were also rarely detected in Fra-2 Tg mice. In summary, the Fra-2 Tg mouse model recapitulates many aspects of human lung pathology, including inflammation, vascular remodelling and fibrosis, but does not recapitulate all the abnormalities of human disease.

Nevertheless, the Fra-2 Tg mouse model can serve as a valuable model to elucidate key pathological mechanisms of lung remodelling and to identify potential therapeutic approaches. The advantage of the Fra-2 Tg mouse lies in its capacity to spontaneously develop phenotypes without the need for disease induction. This model effectively replicates several elements of human SSc-ILD pathology, such as inflammation, vascular remodeling, and lung fibrosis. The complexity of phenotypes and systemic implications in this mouse model closely resemble those observed in humans, thereby aiding in the validation of potential therapeutic targets. The Fra-2 Tg mouse model presents a limitation due to the widespread overexpression of Fra-2 across various cell types, leading to a complex phenotype that impacts multiple tissues. This lack of cell specificity raises uncertainty regarding whether the pathological effects of Fra-2 are directly or indirectly influenced by its presence. Moreover, the high phenotypic variability observed in Fra-2 Tg mice may be attributed to modulation of the inserted transgene by the genetic environment. Finally, because Fra-2 Tg mice do not develop autoimmunity, they can only be considered as a model of the microvascular and fibrotic disease manifestations of SSc. Lack of the Fra-2 gene in mice is lethal, and most studies to date on the role of Fra-2 in lung disease have used Fra-2 Tg mice, with only a few studies using tissue specific expression of Fra-2. However, with the development of science and technology, the selection of tissue specific Fra-2 knockout/overexpressing mice in future experiments may avoid some of the limitations of the Fra-2 Tg mouse model.

In general, the authors should be more critical to their conclusions. Showing that Fra2 is overexpressed in a given condition does not mean that it plays an important role in this condition.

The author’s answer: Thank you very much for your suggestion. We agree with this suggestion. Fra-2 plays an important role not only in the development of lung disease but also in physiological processes such as alveolar regeneration and epidermal differentiation. We have revised the text to include a discussion of the important role of Fra-2 in alveolar regeneration and epidermal differentiation. Details are as follows:

In conclusion, Fra-2 is crucial not only in the progression of respiratory conditions (Table 2) like COPD, pulmonary fibrosis, asthma, and lung tumours but also in various other diseases including cardiovascular issues [136-138], inflammatory disorders, and autoimmune conditions. Additionally, Fra-2 plays a significant role in physiological processes like alveolar regeneration and epidermal differentiation. The ability of adult stem cells to acquire cellular plasticity is essential for prompt tissue regeneration following injury. Choi et al. [139] discovered that in the process of alveolar regeneration, IL-1β signaling regulates Jag1/2 expression in ciliated cells, which then inhibits Notch signaling in secretory cells. This pathway is responsible for driving reprogramming and facilitating differentiation plasticity. Subsequent research revealed that the IL-1β-Notch-Fra-2 pathway is crucial in the transformation of secretory cells into AT2 cells in the process of repairing damage.

Also, since Fra2 is considered as an important transcription factor, providing examples of genes whose expression in regulated by Fra2 would be important to comment.

The author’s answer: Thank you very much for your suggestion. We agree with this suggestion and have added Table 1 to the paper listing the target genes directly regulated by Fra-2.

Thank you very much for your attention and time. Look forward to hearing from you.

Thank you and best regards.

Yours sincerely,

Shuping Zheng

Department of Respiratory and Critical Care Medicine, The Second Affiliated Hospital of Xi'an Jiaotong University, Xi'an, 710004, Shaanxi Province, PR China.

*Corresponding author: Yun Liu, E-mail: [email protected]

Reviewer 2 Report

Comments and Suggestions for Authors

The authors have submitted a comprehensive review of Fra-2 in respiratory diseases, covering over 4 decades of work. The authors will have to generate more figures and tables to better summarise their work and engage readers.

The authors can improve figure 1 with the relative expression of Fra-2 in the various organs if that information is available. This will highlight the abundance of Fra-2 in the lungs.

The authors should include a figure detailing the signaling pathways (or at least what is currently known) upstream and downstream of Fra-2, highlighting potential crosstalks with known pathogenic pathways in respiratory diseases. This will provide the readers a broad summary of the evidence.

The authors should include a table with key papers that implicate Fra-2 role in various respiratory diseases to aid the readers in consolidating the information in the paper.

The references in this review are mostly dated more than 5 years ago. There seem to be a scarcity in recent years on papers involving Fra-2 in respiratory diseases. The authors mentioned that targeting Fra-2 is challenging, but they can better highlight other known mediators of Fra-2 which may be interest for targeting and comment on what is hindering the field and how the field can move forward.

The authors should move the paragraph covering the "Regulation of Fra-2" to before its role in respiratory diseases. This will help with the flow.

The authors should include some preclinical drug development work (if available) done on Fra-2 or its related targets.

Author Response

Response to Reviewer 2 Comments

Dear reviewer,

Thank you very much for your comments and professional advice. These opinions help to improve academic rigor of our article. Based on your suggestion and request, we have made corrected modifications on the revised manuscript. Meanwhile, the manuscript had be reviewed and edited by language services of AJE. We hope that our work can be improved again. Furthermore, we would like to show the details as follows:

Reviewer 2

Comments and Suggestions for Authors

The authors have submitted a comprehensive review of Fra-2 in respiratory diseases, covering over 4 decades of work. The authors will have to generate more figures and tables to better summarise their work and engage readers.

The author’s answer: Thank you very much for your suggestion. We agree with this suggestion and have carefully drawn 2 diagrams and 3 tables to categorise and simplify the content of the article for the convenience of our readers.

The authors can improve figure 1 with the relative expression of Fra-2 in the various organs if that information is available. This will highlight the abundance of Fra-2 in the lungs.

The author’s answer: Thank you very much for your suggestion. We agree with your suggestion and have collected the expression data of Fra-2 protein in different organs of the human body from the Human Protein Atlas database (https://www.proteinatlas.org/) and replotted Figure 1.

The authors should include a figure detailing the signaling pathways (or at least what is currently known) upstream and downstream of Fra-2, highlighting potential crosstalks with known pathogenic pathways in respiratory diseases. This will provide the readers a broad summary of the evidence.

The author’s answer: Thank you very much for your suggestion. We agree with your suggestion. By compiling previous literature, we have mapped Figure 2, which details the currently known signalling pathways upstream and downstream of Fra-2, highlighting potential links to known pathogenic pathways in respiratory disease, to provide the reader with a broad summary of the evidence.

The authors should include a table with key papers that implicate Fra-2 role in various respiratory diseases to aid the readers in consolidating the information in the paper.

The author’s answer: Thank you very much for your suggestion. We agree with your suggestion and have listed the main papers describing the role of Fra-2 in different respiratory diseases in Table 2 to help the reader to integrate the information in the papers.

The references in this review are mostly dated more than 5 years ago. There seem to be a scarcity in recent years on papers involving Fra-2 in respiratory diseases. The authors mentioned that targeting Fra-2 is challenging, but they can better highlight other known mediators of Fra-2 which may be interest for targeting and comment on what is hindering the field and how the field can move forward.

The author’s answer: Thank you for your suggestion. We agree with your suggestion. This review is a comprehensive screening of articles from the last 10 years of research on Fra-2 and respiratory disease.

The authors should move the paragraph covering the "Regulation of Fra-2" to before its role in respiratory diseases. This will help with the flow.

The author’s answer: Thank you very much for your suggestion. We agree with your suggestion and have moved the paragraph 'Regulation of Fra-2' to precede its role in respiratory disease.

The authors should include some preclinical drug development work (if available) done on Fra-2 or its related targets.

The author’s answer: Thank you very much for your suggestion. We agree with your suggestion and have prepared Table 3 to provide a summary of some of the preclinical drug development work that has been done against Fra-2 or its related targets.

Thank you very much for your attention and time. Look forward to hearing from you.

Thank you and best regards.

Yours sincerely,

Shuping Zheng

Department of Respiratory and Critical Care Medicine, The Second Affiliated Hospital of Xi'an Jiaotong University, Xi'an, 710004, Shaanxi Province, PR China.

*Corresponding author: Yun Liu, E-mail: [email protected]

Round 2

Reviewer 1 Report

Comments and Suggestions for Authors

The authors responded satisfactorily to my comments

Author Response

Dear Reviewer:

On behalf of my co-authors, we greatly appreciate your positive and constructive comments and suggestions on our manuscript entitled "Progress in the study of Fra-2 in respiratory diseases" (IJMS-3011622). Thank you again for your comments and suggestions. We hope that the revised manuscript will be accepted by the INTERNATIONAL JOURNAL OF MOLECULAR SCIENCES. If further revisions are needed, please contact me. If you have any questions, please don't hesitate to contact me at the address below.

Thank you and best regards.

Yours sincerely,

Shuping Zheng

Department of Respiratory and Critical Care Medicine, The Second Affiliated Hospital of Xi'an Jiaotong University, Xi'an, 710004, Shaanxi Province, PR China.

*Corresponding author: Yun Liu, e-mail: [email protected]

Reviewer 2 Report

Comments and Suggestions for Authors

I was unable to view the diagrams or tables which the authors mentioned in the current manuscript.

Author Response

Response to Reviewer 2 Comments

Dear reviewer,

Thank you very much for your comments and professional advice. These opinions help to improve academic rigor of our article. Based on your suggestion and request, we have made corrected modifications on the revised manuscript. Meanwhile, the manuscript had be reviewed and edited by language services of AJE. We hope that our work can be improved again. Furthermore, we would like to show the details as follows:

Reviewer 2

Comments and Suggestions for Authors

I was unable to view the diagrams or tables which the authors mentioned in the current manuscript.

The author’s answer: Thank you very much for your suggestion. I'm very sorry again for the mistake of not showing the tables and figures in the main text, as last time I uploaded the revisions separately from the tables and figures. This time I have typeset the body of the revision with the table and figure and uploaded them together. We have carefully drawn 2 diagrams and 3 tables to categorise and simplify the content of the article for the convenience of our readers. The details are as follows:

Round 3

Reviewer 2 Report

Comments and Suggestions for Authors

The authors have addressed all my comments, I have no further comment.